# Non-Destructive Characterization of Cured-in-Place Pipe Defects

**DOI:** 10.3390/ma16247570

**Published:** 2023-12-08

**Authors:** Richard Dvořák, Luboš Jakubka, Libor Topolář, Martyna Rabenda, Artur Wirowski, Jan Puchýř, Ivo Kusák, Luboš Pazdera

**Affiliations:** 1Institute of Physics, Faculty of Civil Engineering, Brno University of Technology, 60190 Brno-střed, Czech Republic; lubos.jakubka@vut.cz (L.J.); libor.topolar@vut.cz (L.T.); jan.puchyr@vut.cz (J.P.); ivo.kusak@vut.cz (I.K.); lubos.pazdera@vut.cz (L.P.); 2Department of Concrete Structures, Faculty of Civil Engineering, Architecture and Environmental Engineering, Lodz University of Technology, 90-924 Lodz, Poland; martyna.rabenda@p.lodz.pl; 3Department of Structural Mechanics, Faculty of Civil Engineering, Architecture and Environmental Engineering, Lodz University of Technology, 90-924 Lodz, Poland; artur.wirowski@p.lodz.pl

**Keywords:** non-destructive testing, machine learning, retrofitting, cured-in-place pipes, polymers, pipe defects

## Abstract

Sewage and water networks are crucial infrastructures of modern urban society. The uninterrupted functionality of these networks is paramount, necessitating regular maintenance and rehabilitation. In densely populated urban areas, trenchless methods, particularly those employing cured-in-place pipe technology, have emerged as the most cost-efficient approach for network rehabilitation. Common diagnostic methods for assessing pipe conditions, whether original or retrofitted with-cured-in-place pipes, typically include camera examination or laser scans, and are limited in material characterization. This study introduces three innovative methods for characterizing critical aspects of pipe conditions. The impact-echo method, ground-penetrating radar, and impedance spectroscopy address the challenges posed by polymer liners and offer enhanced accuracy in defect detection. These methods enable the characterization of delamination, identification of caverns behind cured-in-place pipes, and evaluation of overall pipe health. A machine learning algorithm using deep learning on images acquired from impact-echo signals using continuous wavelet transformation is presented to characterize defects. The aim is to compare traditional machine learning and deep learning methods to characterize selected pipe defects. The measurement conducted with ground-penetrating radar is depicted, employing a heuristic algorithm to estimate caverns behind the tested polymer composites. This study also presents results obtained through impedance spectroscopy, employed to characterize the delamination of polymer liners caused by uneven curing. A comparative analysis of these methods is conducted, assessing the accuracy by comparing the known positions of defects with their predicted characteristics based on laboratory measurements.

## 1. Introduction

Retrofitting a pipe network is the key to ensuring urban and industrial infrastructure’s long-term operational and reliable functionality. Urban and industrial areas heavily rely on water and sewage networks, constituting a civilized infrastructure component [1]. The reliability and durability of these networks are critical, given their interdependence with other systems. However, the retrofitting process of these networks can be intricate and multifaceted [2]. Over time, pipe networks experience deterioration, leading to various defects contingent upon factors such as the construction type of the pipes, environmental conditions, and the materials used. Urban environments often feature extensive pipe networks that demand substantial financial investments. Two primary approaches emerge when assessing civil engineering options: trench pipe retrofitting and trenchless retrofitting processes [3].

In [4], a team led by Travnicek et al. compared the root causes of wastewater treatment plants (WWTPs) shutdowns and accidents. The cited study was conducted using data from public databases (ARIA Database and French, 2020, https://www.aria.developpement-durable.gouv.fr/the-barpi/the-aria-database/?lang=en (accessed on 2 October 2023); eMARS, 2023, https://emars.jrc.ec.europa.eu/EN/emars/content (accessed on 2 October 2023)), in the Czech Environmental Inspectorate database (CEI, 2023), https://www.cizp.cz/en (accessed on 2 October 2023), as well as in professional publications and journals. The comparision of type of defect on WWTP are presented on Figure 1.

Apart from the wastewater itself (categories ‘Tank’ and ‘Entire WWTP’), the highest cause of accidents lies in ‘Pump’ at 11% or ‘Pipeline’ at 10%, which shows that outside of the WWTP, the pipeline defects or overall deterioration is the main cause for the WWTP shutdowns.

For the repairs of the pipes, trenching, characterized by simplicity, cost-efficiency per meter, and time effectiveness, is commonly employed in open areas. In contrast, industrial areas may witness an escalation in the retrofitting process cost, time, and complexity, rendering the standard trench and pipe replacement method impractical [1,5]. This issue is particularly pronounced in urban settings, where property rights can result in protracted administrative procedures for retrofitting. Moreover, specific networks may not be amenable to modification without extended technological disruptions or water service interruptions [6]. These challenges diminish the convenience of trench retrofitting, making trenchless retrofitting processes more viable from both an efficiency and economic perspective [7].

Trenchless technology encompasses a range of techniques aimed at installing, replacing, or renewing underground utilities with minimal excavation and surface disruption. These techniques have been successfully applied to underground utilities, including water, sewer, gas, industrial pipelines, electrical conduits, and fiber optics [8]. Trenchless technology is an attractive construction option, particularly in densely urbanized areas with heavy vehicular and pedestrian traffic and many existing underground utilities. It benefits crossing roadways, transportation corridors, rivers, waterways, environmentally sensitive areas, and locations where surface access may be constrained due to existing structures or vegetation [9,10]. Trenchless techniques often represent the sole practical construction approach, being cost-effective and minimally disruptive. This approach addresses the perennial issues associated with road works, including traffic congestion, fuel consumption, environmental pollution, accidents, structural damage, and inconvenience to the public [11].

In the realm of trenchless technology, one notable application area is the rehabilitation of underground pipes. Trenchless technologies are widely recognized for their cost competitiveness in this domain. Many utility pipelines, particularly sewage systems, suffer from corrosion due to modern effluents, overloading, loss of capacity, variations in material, wall thickness issues, rehabilitation requirements, and the need for minimal service shutdown times [12,13].

Various rehabilitation techniques exist, including cured-in-place pipes, close-fit lining, slip lining, spray lining, and other localized repair methods. For instance, CIPP (cured-in-place pipes) involves inserting a fabric impregnated with polyester or epoxy resin into the defective pipe, then inflating and curing it to the inner surface of the host pipe. This method is adaptable to varying pipe sizes and is commonly used for rehabilitating gravity sewers with no loss of capacity [14].

An example of retrofitting action of sewage pipes in city park Lužánky in Brno city is shown in Figure 2. For the CIPP technology, a starting shaft is dug up, and then the rest of the sewage pipe is retrofitted with a cured-in-place pipe without digging the whole segment of the network.

Additionally, cement or resin spray linings have extensive application in water pipelines but require careful selection due to potential solvent and residue release. Slip lining involves placing a new pipe within an existing one and grouting the annular space between them, although with some reduction in capacity. This method can now employ polyethylene to minimize the annular space [14,15].

Moreover, modern robots equipped with CCTV cameras are deployed for cleaning, preparing, and filling cracks and voids with epoxy mortar, offering a cost-effective solution for isolated problems in structurally sound pipelines. The ease of equipment transport and mobilization further enhances its appeal [15].

Spiral wound lining is another technique in which a PVC strip is fed through limited access points into the defective pipe and wound helically against the pipe wall using a winding machine operated from within the line. This approach is advantageous for emergency repairs and enhancing the strength of weakened pipelines [14,15].

Research in in-line pipeline inspection has intensified, with a focus on non-destructive testing (NDT) methods for discontinuity detection and safety assessment. Conventional NDT techniques, such as radiographic, penetrant, ultrasonic, visual, eddy current, and magnetic particle testing, have been effective. Still, their suitability depends on specific requirements and the nature of defects. Tailoring detection methods to particular needs is crucial, considering pipeline characteristics and limitations [16,17].

The above scenarios are particularly suitable for metallic pipes. The scientific and engineering communities have developed techniques for detecting defects in operational pipelines, classified into three main categories for pipeline maintenance [18]. These encompass locating pipelines and underground facilities, identifying excavation damage and encroachments to the right of way, leak detection, and damage mitigation [19]. Excavation damage is a significant threat to gas pipelines, often indicated by leak detection. With recent advancements like acoustic and ground-penetrating radar technologies, developing specialized electromagnetic locators for metallic pipeline positioning is essential to protect the pipeline right of way. Numerous in-line inspection methods based on NDT have been developed internationally as the most effective approach to detect and locate pipeline defects and quantify stress [20].

Risk assessment and information management technologies encompass data visualization, asset tracking, geographical information systems, risk assessment, response awareness, network, and physical security. These technologies assist in identifying high-risk pipelines for repair or replacement. The current research focuses on metallic pipes, mainly used for strategic energy products such as oil, gas, etc. [21].

The inspection strategies for metallic pipes are as follows: ultrasonic inspection, acoustic emission inspection, eddy current technique, magnetic flux leakage inspection, eddy current pulsed thermography, magnetic Barkhausen noise, radiography testing, NDT methods like penetrating testing, magnetic particle testing, and visual testing are standard in the industry, but inspecting buried pipelines is challenging and costly [22]. In-line inspection devices, including oil and natural gas lines, water pipes, and submarine systems, are widely used to inspect pipelines. These smart devices collect physical data about pipeline integrity inside the pipe. However, only ultrasonic, electromagnetic, magnetic flux leakage, and eddy current testing methods have successfully developed tools for in-line pipeline inspection due to volume and length limitations [16,21].

In the field of sewage pipelines, the situation is different. Waste pipes are primarily made up of concrete or formerly ceramic pipes. In this case, complications arise regarding the accurate detection and location of defects, as these construction materials are somewhat more complicated regarding wave propagation. In the case of concrete waste pipes, inspection tools similar to those used in civil engineering diagnostics can be used. These tools include visual inspection, ultrasonic testing, resonance inspection, ground-penetrating radar, etc. [23].

### Cured-in-Place Pipe Testing

The rehabilitation of underground sewage and water pipelines represents a critical component of modern infrastructure management, ensuring these essential systems’ continued reliability and functionality. Among the various techniques employed for pipe rehabilitation, the CIPP method has gained prominence for its ability to efficiently restore the structural integrity of aging pipelines while minimizing the disruptive impacts of traditional excavation and replacement [24].

Central to the success of the CIPP method is the integrity and performance of the polymer-based liners used to create new pipe structures within the existing infrastructure. Polymers, including epoxy resins and other composite materials, have become pivotal in CIPP applications due to their high tensile strength, corrosion resistance, and durability. These attributes are essential for ensuring the longevity and reliability of the rehabilitated pipes [25].

The typical process of retrofitting sewage pipes by the CIPP method can be separated into several steps. It begins with an initial inspection to assess the condition of the existing pipe. Following this, cleaning and preparation are carried out to ensure the pipe’s interior is free from debris and obstructions. Measurements are taken to determine the appropriate liner size and material, with epoxy resin commonly used. The selected liner is impregnated with resin, inserted into the existing pipe, and inflated to conform to its shape. The curing process ensures the liner hardens and forms a new pipe within the old one [26]. To track the curing quality, the liners often have temperature sensors to record the temperature along the retrofitted segment. The record of the temperature gradient is then stored for later guarantee checks. Once completed, cooling and a final inspection are performed, followed by re-connection and testing of the rehabilitated pipe to confirm its functionality. After successful testing, the project is considered complete, providing a cost-effective and trenchless solution for rehabilitating underground pipes. The process is illustrated by the scheme in Figure 3.

To better describe the process of liner filling the host pipe, we present the illustration in Figure 4.

The structure of liners can be more separated into these parts:Epoxy Resin: This is a thermosetting polymer that is initially in a liquid state. It serves as the primary material used to impregnate the liner fabric. Epoxy resin provides the structural strength and rigidity to the cured liner.Felt or Non-Woven Fabric: The liner fabric, often made of felt or non-woven material (such as polyester or fiberglass), is impregnated with epoxy resin. This fabric serves as the reinforcement layer that provides the liner’s structural integrity and strength.Water-Tight Polymer Layer: On the inside of the liner, a water-tight polymer layer is typically applied. This layer acts as a barrier to prevent water infiltration into the cured liner. It enhances the liner’s resistance to corrosion and helps maintain hydraulic efficiency.Inflation Tube: An inflation tube or bladder is inserted into the liner before installation. This tube allows for the liner’s expansion and ensures it conforms to the shape of the host pipe during the curing process.Catalysts and Accelerators: Depending on the epoxy resin used, catalysts and accelerators may be added to control the curing time and temperature. These additives help initiate the chemical reaction that transforms the liquid resin into a solid, hardened liner.Release Liner: A release liner or backing material is often used to protect the resin-saturated fabric during transport and installation. It is removed before the liner is installed in the host pipe.

However, the successful implementation of CIPP and the long-term assessment of liner performance have introduced a unique challenge: measuring and evaluating the acoustic properties of polymer-based materials with significantly different acoustic impedance compared to the original pipe materials, such as concrete or clay. Acoustic impedance, a fundamental property of materials affecting the propagation of sound waves [27], plays a critical role in detecting and characterizing defects, voids, or irregularities within pipe structures. The disparity in acoustic impedance between the polymer liners and the surrounding materials can complicate the application of NDT methods.

The primary approach in sewage pipe diagnostics involves visual inspections, often using CCTV surveys, to assess their condition. These inspections help identify defects, plan maintenance, and predict remaining lifespans. However, this approach has limitations, including the potential for missing hidden defects, accuracy dependence on equipment and expertise, and challenges in identifying underlying causes. It can also be time-consuming and costly for extensive networks, and predictive modeling may introduce uncertainties. Emerging technologies and data analytics are being explored to address these limitations and improve accuracy and efficiency in sewage pipe diagnostics [28,29,30]. The example of the most usual defects within the CIPP method is shown in Table A1.

Apart from the standard methods mentioned above, new methods are also being developed. The case study of a novel test procedure for the Balloon Pressure test was presented by Ferran Gras-Travesset et al. [31]. The designed procedure is based on placing an inflatable balloon inside the CIPP segment and loading the CIPP inner hole with the pressurized prototype balloon device. The device is meant to test the selected segments of the pipe network and measure the tensile strength of the whole ring segment.

This scientific paper explores the innovative use of the IE method as a promising approach to address the acoustic impedance challenge in CIPP applications. The IE method, which relies on generating and analyzing stress waves to assess the internal condition of materials, offers the potential to overcome the limitations posed by polymer liners’ unique acoustic properties. Through advancements in IE testing techniques and signal analysis, this research aims to enhance the accuracy and reliability of defect detection, structural assessment, and quality control within CIPP-rehabilitated pipes.

In the following sections, we delve into the principles of the IE method, its application in CIPP testing, and the development of tailored methodologies to effectively address the acoustic impedance mismatch between polymer liners and the surrounding pipe materials. Ultimately, this research seeks to contribute valuable insights and methodologies for the improved evaluation of CIPP rehabilitation projects, ensuring the continued integrity and functionality of vital underground pipelines.

## 2. Materials and Methods

To demonstrate the capability of the selected methods, a test board made from epoxy resin and polyester (PES) non-woven fabric were made in diameters of 1.1×1.3 m. Then, different numbers of layers of liners were stacked to simulate different thicknesses of CIPP. The selected boards were then divided into 100×100 mm test patches, used as reference points for testing by different methods. The boards were created with artificial defects of delamination in the selected areas as seen in Figure 5.

To have a tight adhesion between the liners, the boards were vacuumed by a vacuum pump. At the selected region, the boards were intentionally let free so air could form bubbles between the liners. These defects are well described in Section 3.2. They can be fitted in the types 1 ‘Liner Delamination’ and 2 ‘Liner Wrinkles or Folds’ from the table of typical defects in Table A1.

For the input materials, epoxy resin cured at ambient temperature 20 °C and as a liner manually stacked layers of non-woven poly-ethylene fabric were used. A manufacturer of Epoxy resin in INCHEMIE Technology s.r.o. from Olomouc, Czech Republic, and the resin has code IN-EPOX 6040. For liner layers a PES Felt PPX from manufacturer AEGION Northamptonshire, UK was used. A compacting roller homogenized the layers. The detailed description of input properties is shown in Table 1.

The removal procedure of specimens in situ is limited to the original shape of the host pipes, which in most cases does not allow producing flat specimens. Due to this fact, the specimens are always in semi-circular shape. To artificially produce the defect of delamination, we needed to select the flat shape of boards.

### 2.1. Experiment Setup

The boards were tested with two sets of defects. Boards A and C had different sets, and Board B was measured with the set of Defects 1 and 2. Board A was subsequently cut into specimens 100×100 mm for measuring the apparent density and dielectric constant $\varepsilon^{\prime }$. A more detailed description of tested boards is presented in Table 2.

To precisely track the position of defects in the sand, an image recognition technique was used [32] where an image of the board and an image of defects without the board were fit onto each other using an estimator application in MATLAB R2023a. This was carried out for each tested board to obtain precise coordinates of defects either for ground-penetrating radar or impact-echo testing. An example of this procedure is shown in Figure 6.

An overview of all boards is presented in Figure 7. To localize caverns and different materials below the CIPP boards, plastic PVC pipes of diameters ⌀ 100 and 150 mm were used. To test the location of different materials, a plastic bottle and a metal aluminum were placed in Defect Set 2 (blue and white color).

The acoustic properties of CIPP being pushed against the host pipe by the curing medium influence the resulting inner rigidity of the cured composite, which leads to different resonance frequencies along the cross-section of the pipe. The resonance frequency differs because the whole body of CIPP is at some contact points pushed against the host pipe with a bigger force than in the upper part of CIPP, which hangs with less resisting force from the host pipe. The defects within such specimens can be classified, but the resulting analysis only applies to the cross-section type or diameter. The rectangular board shape for the specimen was selected to assess the accuracy and stability of the measured data presented in the paper. This enables the simulation of more types of defects, with a combination of placing the bodies and volumes with different electromagnetic properties (sand, polymers, steel, aluminum, air).

Such a setup aims to assess the capability of impact-echo or ground-penetrating radar to localize caverns and different materials below the CIPP. Due to the artificial wrapping shown in Figure 5, the IE method, impedance spectroscopy, and apparent density are compared to describe the caverns and defects within the CIPP boards.

After the manufacturing of the specimens, they were placed on the sand bed, shown in Figure 6. First, the board was measured without the defects, then with the defects using the ground-penetrating radar, followed by impact-echo testing. When the first board was tested by both methods, it was changed for the next board. The order of boards was A→B (change of defects from 1 to 2) B→C. After A was switched to B, the Board A was cut to 100×100 patches for measurement of apparent density and electric impedance.

### 2.2. Impact-Echo

The impact-echo method is a non-destructive testing technique utilized to assess the integrity and internal condition of materials, particularly in civil engineering and infrastructure inspections. IE relies on generating and analyzing stress waves or acoustic signals within the material being tested [33,34].

Within IE testing, the behavior of waves at material interfaces plays a pivotal role. Snell’s law, a fundamental principle in wave physics, guides our understanding of how waves interact when transitioning from one medium to another, each with a different acoustic impedance. In IE assessments, waves can take various forms, including longitudinal waves (P-waves), shear waves (S-waves), and surface waves (Rayleigh waves). P-waves are ideal for determining the material’s depth and thickness, while S-waves are valuable for detecting internal defects. Rayleigh waves, on the other hand, provide insights into surface conditions.

Traditionally, piezoceramic accelerometers are utilized in IE testing to capture stress waves. However, alternative sensors, such as microphones, can also be adapted. When employing a microphone, it is crucial to ensure that its frequency response aligns with the range of stress wave frequencies generated by the impact source and echoes. The adaptability of the microphone expands the applicability of IE testing, offering versatility in various NDT applications, avoiding the demanding sensor attaching process to the testing elements. The scheme of testing is shown in Figure 8.

In IE testing, several crucial parameters and data acquisition (DAQ) settings are employed to ensure accurate and reliable results. The sampling frequency was set to 200 kHz with 24-bit resolution. For measuring the echo signals, a Presonus PRM1 was used [35]. This microphone has a dynamic range of 70 dB with a max SPL of 132 dB, which ensures that fast changes from silence to impact signals are recorded without overflowing the maximum range. For recording the signals, a Scarlett 3rd-Gen audio board was used [36]. A steel hammer with a ball tip of a diameter of 16 mm was used as an impact source. The signals were recorded using a rising-edge threshold set to 0.02 V with a 0.05 s pre-trigger. The total length of the signal was set to 0.5 s.

Typically, for assessing the recorded signals, an analysis in frequency region using fast Fourier transformation is used. With this analysis, we can get the main dominant frequencies and their harmonic frequencies, which is the main approach in interpreting IE measurements. The advantage is relatively fast computational time with an easy-to-read amplitude output at different frequencies. The downside is that some frequencies occur in the recorded signal at a specific moment of the signal, which can indicate the inner heterogeneity or the state of the defect, which is otherwise not recognizable with fast Fourier transformation (FFT). The FFT, PWT, and CWT comparison is shown in Figure 9.

This transformation allows the signal to be decomposed into the individual frequencies of which it is composed. It is an approximate estimation of the individual frequencies ω of a short time period of the signal $t_{0}$. The FFT can presented using Formula (Equation 1):(1)$$F\left(\omega \right)=\frac{1}{\sqrt{2\pi }}\int_{0}^{\infty }\;f\left(t\right)e^{-i\omega t}\;dt$$

Apart from the FFT, a power spectrum, if the signal is also being used, can give a better understanding of the noised signal [37]. Alternatively, continuous wavelet transformation (CWT) presents an alternative to FFT, offering specific advantages when dealing with non-stationary signals commonly encountered in IE testing. Wavelets can simultaneously capture time and frequency information, rendering them suitable for identifying transient features and discontinuities in the data. The CWT can be represented by Formulas (Equation 2) and (Equation 3) [38]:(2)$$WT\left(\tau ,S\right)=\int_{0}^{\infty }\;x\left(t\right)\Psi_{\tau ,s}\left(t\right)dt$$
(3)$$\Psi_{S,\tau }\left(t\right)=\frac{1}{\sqrt{\left|S\right|}}\Psi \left(\frac{t-\tau }{S}\right)$$
where τ is the time shift; s is the scale that is related to the frequency; Ψ is the transformation function, which is also called the basis or mother function; x(t) is the analyzed signal; and $WT\left(\tau ,s\right)$ is the resulting time–frequency transformation. Using the CWT approach, a scalogram can be retrieved, a 2D representation of the given signal, as seen in Figure 9. Apart from the signal and frequency spectrum, this 2D matrix is used to classify the measured signals based on their selected class. A similar approach can be seen in the most recent publications [39,40,41]. In the presented paper, a Python library for fast continuous wavelet transformation was used and published by Arts, Lukas P. A. [42].

To acquire the IE signals, the testing procedure of the whole board has been designed. Each board was divided into 10×12 testing points. From each testing point, 3 signals were acquired. Firstly, the bottom row was measured, after that second row. The measuring of the points was oriented from left to right. Because the signals were measured using a rising-edge threshold, some of these signals had a defect, or were noised beyond the recognition. These signals were filtered out by localizing the start of the signal and the overall signal-to-noise ratio of the signal. In total, 1352 signals were recorded from all 4 specimens. In total, 50 signals were not recorded properly and were removed from the dataset, which represents 3.7%.

In the past work of the team of authors impact-echo signals have been interpreted using different machine learning approaches [43], where a machine learning (ML) algorithm was used in order to classify the temperature degradation of concrete specimens. To find the caverns within the concrete elements, the work of Sattar Dorafshan et al. can be mentioned [44]. In this example, a deep learning (DL) model was used to localize caverns within concrete test specimens. The approach was using the DL algorithm with ResNet and GoogleNet neural network architecture. To increase the classification accuracy, the continuous wavelet transformation was also used, which has shown higher accuracy compared to the standard frequency spectrum analysis. These examples show the versatility of the IE method, which due to its simplicity can generate interesting results in combination with machine learning and deep learning methods. However, it has a limited application due to the need for training data for effective use in specific applications. Worth mentioning is the model presented in [43], where the shape of the tested body is limited to standardized diameters. In such cases, a high accuracy, together with reproducibility, is expected. If the model is used on the whole structure, much bigger training datasets are needed. Another example of the usage of IE on the concrete sewage pipe is the paper published by Seokmin Song et al. [45], where the IE method was used to detect caverns behind the concrete sewage pipe. In such examples, it is important to note that concrete has much higher acoustic impedance, and this localization is less difficult than in the case of semi-cured CIPP composite with the presence of defects.

### 2.3. Ground-Penetrating Radar

Ground-penetrating radar (GPR) is a non-invasive geophysical technology used to explore and map subsurface features and structures in the ground, such as soil, rocks, water, and buried objects. It is widely used in various fields, including archaeology, geology, civil engineering, environmental science, and utility detection [46].

GPR operates on the principle of sending electromagnetic pulses into the examined object and measuring the time it takes for these pulses to bounce back to the surface after encountering subsurface materials with different dielectric properties. The dielectric properties of materials determine how well they conduct electrical energy, and this property varies between other substances.

The GPR system consists of a transmitter and a receiver. The transmitter emits short bursts of electromagnetic waves, typically in the microwave or radio frequency range. The frequency used depends on the specific application and depth of penetration required. When electromagnetic waves encounter subsurface materials with contrasting dielectric properties (e.g., soil and buried objects), some waves are reflected to the surface, while others continue to penetrate deeper. The reflected waves carry information about the composition and depth of the subsurface features they encounter. The receiver records the time it takes for the reflected waves to return to the surface. Geophysicists can create a detailed subsurface profile by analyzing these reflections’ time delays and amplitudes.

The collected data are processed and displayed as radargrams or depth profiles. Geoscientists interpret these images to identify and characterize subsurface features. Different materials and objects can produce distinct reflections in the data, allowing the identification of buried pipes, archaeological artefacts, geological strata, groundwater tables, and more. The depth of penetration and resolution of GPR depends on several factors, including the frequency of the radar waves, the dielectric properties of the materials, and the equipment used. Higher frequencies provide better resolution but more shallow penetration, while lower frequencies penetrate deeper but offer lower resolution [47].

We use the GPR from the company Ingegneria Dei Sistemi, type RIS PLUS S13633, with an HFT2000 probe and a sampling frequency of 5 GHz. As an evaluation software, we use Greed HD.v02.02 The used ground-penetrating radar illustration is shown in Figure 10.

Because the measured specimens were in the shape of matrix points as presented in Figure 6, the same number of B-scans was taken in the *x*-axis and the *y*-axis, and for each board is then taken a sum of 20 scans. The process of scanning is illustrated in Figure 11. The probe has a tracking wheel, which stores the linear coordinates of each B-scan taken.

The ground-penetrating radar is an overall versatile tool to document the underground cavities, which finds usage in archaeology [48], geology mapping [49], urban planning [50], diagnostics in the civil engineering [51], or scanning subsurface of different planets [52]. In civil engineering, the most common usage is to localize and characterize re-bars for the purpose of core bore specimens or probes on the bridge bodies [53].

### 2.4. Impedance Spectroscopy

The ROHDE&SCHWARZ ZNC Vector Network Analyzer has a bidirectional test set for measuring all four S-parameters of active and passive DUTs (devices under test). Operating from 9 kHz to 3 GHz, the network analyzer targets mobile radio and electronic goods applications. The R&SZNC is the right choice for developing, producing, and servicing RF components such as filters and cables. For the frequency range 10 MHz to 3 GHz, it is possible to use the coaxial probe DAK-12 manufactured by SPEAG. In this frequency spectrum, we can measure electrical conductivity and relative permittivity as a function of frequency. The DAK software uses the Virtual Instrument Software v2.0 Architecture (VISA) standard libraries for communication with the VNA (Vector Network Analyzer) via GPIB, RS232, USB, or Ethernet. An example of a probe used on cut-out pieces of measured board A1 is shown in Figure 12.

Impedance spectroscopy in civil engineering is used rather sporadically, mainly because of the preparation of specimens, which requires precise dimensions and a flat contact area between the specimen and measuring probe. The water content inside the specimen also influences the acquired property relative permittivity coefficients. The measured permittivity coefficient under different frequencies refers to the relative permittivity of water, which reaches a value of 1. Apart from this sensitivity to the measured material’s water content, the method’s main advantage is to assess the specimen in volume due to the characteristics of the electromagnetic field generated by the probe [54].

Apart from the relative permittivity comparison, the apparent density was also measured. Because the liner’s internal structure is a non-woven fabric consisting of multiple layers impregnated with epoxy resin, in some cases, an air gap can occur. In such a situation, the surface of the liner can become wavy because of gaps between the layers of the liner wall. To track these changes, Board A was cut into 10×10 cm segments, where width *w*, length *l*, and weight *m* were measured, with maximal thickness $t_{max}$ as well. To determine the distribution of these defects and apparent density, $\rho_{0}$ with unit kg·m^−3^ was then calculated:(4)$$\rho_{0}=\frac{m}{w\;\cdot \;l\;\cdot \;t_{max}}$$
where *m* is in kg, and *w*, *l*, and $t_{max}$ are in m. With this variable, a distribution of defects of delamination and waving inside Board A can be tracked and interpreted.

### 2.5. Machine Learning

Machine learning is a subfield of artificial intelligence (AI) that focuses on developing algorithms and models enabling computers to learn from and make predictions or decisions based on data. It is about creating systems that can improve their performance on a specific task through experience.

Deep learning is a subset of machine learning that involves artificial neural networks (ANNs) with multiple layers (deep neural networks). These networks are designed to automatically learn and represent data through the hierarchical structure of layers. They have become particularly powerful for tasks such as image and speech recognition.

A neural network is a computational model inspired by the structure and functioning of the human brain. It consists of interconnected artificial neuron (nodes) layers that process and transform data. Neural networks are used for various machine learning tasks, including image and speech recognition [55].

Both approaches of ML and DL have the common principle of using the datasets to train the model on the dataset and evaluate its accuracy and reliability. This procedure can be divided into two stages [56]. First, we need to select the proper model, which suits the dataset best. In such cases, the dataset is divided into three groups: train, validation, and test. The training group is used during the learning process to fit the model parameters. The validation set is used for tuning the hyperparameters and, in some publications, is called a test set as well. After the model is trained using the train and validation group, it is used to test predictive accuracy using a test set. In this case, the selected ratios are usually 1/3 of the whole dataset per train, validation, and test set. In the second case, we need to train the selected model using only the train and validation group. In this case, a split ratio is usually around 75:25. However, when working with datasets featuring fewer observations and a higher number of distinct classes, this method may encounter challenges. Specifically, it can be affected by the underrepresentation of certain classes, leading to potential inaccuracies such as false-positive classification or unusually low accuracy. It is important to note that the 75:25 ratio is not a rigid rule, as some research papers opt for different ratios [57,58].

Different approach is the k-folds cross-validation method, as described in [59]. This method involves partitioning the input dataset into subsets, with one subset designated as the test set and the remaining subsets serving as training sets. The classifier then trains a model on the training set and evaluates its accuracy and performance on the test set. This iterative process is repeated multiple times, each time utilizing a different subset for the training and test sets. In the context of this paper, a fold value of five was employed for cross-validation.

To predict if the given signals come from class delamination, cavern, or matrix an image classification is needed. In such cases, the whole image given by its resolution and color channels are fed to the input layer of ANN. Image classification is a computer vision task where a system categorizes or labels images into predefined classes or categories. For example, classifying images of animals into “cats”, “dogs”, or “birds” [60]. The presented paper used ResNet 18 and Resnet 34 neural networks to classify signals acquired by the impact-echo method. A continuous wavelet transformation is obtained from each signal, and the resulting 2D image with 3 RGB channels and resolution of 416×416 pixels is stored. In total, 1352 signals were acquired. The distribution of each signal is shown in Figure 13.

ResNet (residual neural network) is a deep convolutional neural network architecture that has succeeded highly in image classification tasks. Other architectures like VGG, Inception, and AlexNet are also popular for image classification. These networks are pre-trained on large datasets and can be fine-tuned for specific image recognition tasks [61].

The optimizer function is a crucial component in training neural networks. During training, it adjusts the model’s parameters (weights and biases) to minimize the error (loss) between predicted and actual values. Common optimizers include stochastic gradient descent (SGD), Adam, and RMSprop. In this experiment, the stochastic gradient descent (SGD) optimizer for training the neural network model was employed, following the classic approach as outlined by Bottou et al. [62] and further insights by Ruder (2016) [63].

During the training of a neural network, the dataset is typically divided into training and validation sets. Training involves iteratively feeding data into the network and updating its parameters using the optimizer to reduce the loss. Loss is a measure of the error between predicted and actual values. Accuracy is a metric that quantifies how well the model performs on the validation set, often expressed as a percentage [64].

For the training, the ANN used a computer with CUDA support [65]. The CUDA Toolkit provides a comprehensive development environment for C and C++ developers building GPU-accelerated applications, and in the presented paper, was used to accelerate the training process using the PyTorch library [66]. Table 3 shows the parameters used for training the ANN. ResNet18 took 406 s to train, and ResNet36 took 550 s to train.

## 3. Results

The selected boards were placed on the sand bed with defects. The board itself tends to vibrate and move upon the impacts, so before each measurement, it was locked by additional weights, so the board was pressed against the sand badly. The first was a measurement of the board by GPR based on the description in Section 2.3. After the GPR measurement, the impact-echo method was used to test each board at all testing points. For each testing point, three signals were stored. After Board A was completely measured, the board was cut into 100×100 mm samples. These rectangular specimens were then weighted and measured by the impedance spectroscopy method.

### 3.1. Impact-Echo Testing

The traditional impact-echo method is applied to the specimen or construction segment, where the dominant frequency and its harmonic frequencies are assessed. This approach depends on the operator’s experience and skill in interpreting the measured data. In the past years, with the rise of the machine learning and artificial intelligence approach, this procedure has been automated in various ways. The correlation plot of measured data is shown in Figure 14. From this plot, a comparison between the classes can be observed.

For the successful classification of each class, we want the most diverse dataset of the observed defects, their position, and size. In the presented example, the clouds representing the selected features of measured signals are similar. On the plot of crest factor and signal-to-noise ratio (SNR), a shift in SNR in dB between the classes can be distinguished, signaling that this region is diverse enough for a classification model. The description of the used algorithm to retrieve the features from the signals was described in past publications [43,67]. Features consist of statistical variables such as root mean square, signal-to-noise ratio, crest factor, or energy of the signal. The signal features are then enriched by spectral variables, such as a dominant frequency, maximum amplitude, and parameters of dominant peaks such as width or prominence.

Numerous classification algorithms were used to test the standard machine learning approach’s accuracy. The most effective approach is using the classification learning application in the Machine Learning Toolbox within MATLAB software R2023a. This application allows the testing and optimization of multiple classification algorithms, such as decision trees, discriminant analysis, support vector machines, and more. When the presented dataset is fed into this application, a comparison of overall accuracy is retrieved. Because the whole dataset is described by 3 classes and 1146 observations, we used k-fold validation using 5 folds for training instead of the hold-out cross-validation method.

The most successful model was bagged trees, which achieved an accuracy of 75% for all of the specimens. The accuracy of classification for each class was:‘Delam’ was 72%;‘Hole’ was 83%;‘Matrix’ was 71%.

This demonstrates that the standard machine learning approach cannot precisely recognize the classes when the whole dataset of Specimens A1, B1, B2, and C2 is used for the classification. Because the classification of ‘Hole’ is the most challenging, we can compare the total accuracy of the model with its accuracy only for class ‘Hole’. All specimens are then represented by their confusion matrices shown on Figure 15.

From the comparison of the overall accuracy and individual class accuracy, we can see that the highest accuracy can be seen in the case of Defect 2 in Boards B and C (B2, C2) with a total accuracy of 78% and class accuracy 85%. The distribution of Defect 2 in this setup seems more suitable for the classification model than Defect 1, which has the lowest accuracy of 73% for overall accuracy and 83% for class accuracy. Defects 1 at Figure 7a,b are distributed across the whole board, reaching the left, right, top, bottom, and center area, while Defects 2 in Figure 7c,d are placed on the longitudinal line and top side of the board. The distribution of defects affected the signal characteristics, which influenced the ability of the classification model to train the accurate model effectively.

When we look at the whole dataset (A1, B1, B2, C2) and compare it with the trained dataset of the board with different defects (A1, C2), we can see that both groups have similar total and class accuracy.

The fact that traditional machine learning using feature extraction techniques is capable of generating a model of 78% shows limits for a usable model, which could be used in practice, so the deep learning approach is used instead.

The example of representative signals of all three classes is presented in Figure 16.

Each signal is presented in its time and frequency domain with illustrated features used for standard machine learning. Signals are then supplemented by its time–frequency visualization using fast continuous wavelet transformation (CWT) [42]. This 2D image represents the signal and shows more information than the standard frequency spectrum obtained by fast Fourier transformation or as a power spectrum [37]. A deep learning neural network can be trained on CWT diagrams to classify the signals as one of the selected classes.

For training a DL model a hold-out method was used with a ratio of 75:25. To compare different networks, we selected ResNet18 and ResNet34. The training and validation progress is presented in Figure 17. The curves show that both networks are steeply graduating to high training accuracy, with validation accuracy at 90% for ResNet18 and 93% for ResNet34. The situation is rather different if we compare the specific accuracy for classes.

When comparing the accuracy of ML and DL models, interesting differences emerge. The ML model demonstrates similar accuracy in the classification of the ‘hole’ class, achieving an overall accuracy of 83%, whereas the DL model attains 84% (in the case of ResNet34), but with significantly higher accuracy in the rest of the classes. This observation suggests that the DL model may be more reliable in terms of predicting all classes. From the perspective of training and feature extraction, the DL approach is less susceptible to errors stemming from limitations in feature extraction functions. However, the training time for the ML model is 34 s and for DL 7 min respectively, which is 12× longer.

In both cases, the ANN has high accuracy for classes ‘Delam’ and ‘Matrix’, with the fact that ‘Delam’ is not being misclassified as other classes. This makes this class unique in the dataset and means that this type of CIPP defect could be localized by the IE method using the ANN approach. While ResNet18 has a ‘Hole’ class accuracy of 65%, ResNet34 has an accuracy of 84%, with the highest accuracy from all tested standard machine learning models and the ANN. The comparison of confusion matrices of both networks is presented in Figure 18.

The difference in the accuracy between both approaches is substantial, considering that traditional machine learning depends on the feature extraction function. This has to be well-optimized and cover many exceptions, which is often challenging in a rough industry situation. The retrofitting actions are prone to non-orthodox approaches, and the DAQ procedure for IE measurement must be optimized. The successful rate of the feature extraction function can then vary.

On the other hand, the deep learning approach is much easier because pre-trained ANN extracts its features from the input image. Thus, the problems connected to the feature extraction are inside the training procedure itself. This simplifies the whole process of re-training and fine-tuning the network and optimizing function between the epochs. The deep learning approach has a downside due to the high demands on computational power, where a standard office computer is insufficient.

### 3.2. Permittivity and Apparent Density

Board A with Defect 1 (Specimen A1) was cut into 120 samples, each with the row name and column from which they were retrieved. Due to the artificial wrapping defects in the area shown in Figure 5. The typical representative of specimens is shown in Figure 19.

The wrapping defects can be distinguished by delaminating liner internal layers with air gaps in between, as seen in Figure 19b. In some places of the board, the epoxy resin was sucked out completely from the liner, which is shown in Figure 19c, resulting in the lowest measured apparent density. If we compare the overall apparent density $\bar{\rho_{0}}$, we can observe the presence of wrapping Board A, which is shown in Figure 20.

It is important to emphasize that wrapping defects can be, in some cases, distinguished from the surface visually, indicated by waves on the surface, but in the whole area. We have tested the impedance spectroscopy method to overcome the need for cutting out the specimens, which can measure the materials using an electromagnetic impedance probe. This method can measure the relative permittivity $\varepsilon_{r}$ only from one side of the specimens.

The electric impedance $\varepsilon_{r}$, or dielectric constant, is defined by the ratio of the capacitance of a plate capacitor whose dielectric is the substance C to the capacitance of the same capacitor whose dielectric is the vacuum $C_{0}$ (in practice, the capacitance of air is sufficient). The relative permittivity of common substances ranges from 1 to 100.

Electric impedance has been monitored at higher frequencies (10–3000) MHz. The dependence of relative permittivity on frequency is shown in Figure 21a. The waviness of the curves is assumed to be due to systematic error associated with the test equipment since the same pattern is present in all curves.

The results of the measurements and their analysis confirm that changes in the behavior of building materials indicate changes in their structure and, hence, quality.

The measured dependency of electric impedance $\varepsilon^{\prime }$ at different excitation frequency *f* is shown in Figure 21a. From each frequency region, a range of 446 to 522 Hz was selected using the Bayesian optimization function [68], where a control function was correlation coefficient $R^{2}$ between impedance and apparent density.

The overall correlation of both apparent density $\bar{\rho_{0}}$ and electric impedance $\varepsilon^{\prime }$ is not higher than 0.56 after the Bayesian optimization, which shows that impedance in this particular situation corresponds rather to different defects within the board.

These results make the whole method interesting for further research, which could focus on characterizing defects such as incomplete or uneven curing, wrapping, and delamination of the liners.

### 3.3. Ground-Penetrating Radar

Analysis of GPR data is a more difficult issue due to both the nature of the operation of this device and the accuracy of the results obtained. An exemplary volumetric scan of the board is shown in Figure 22.

It is worth pointing out that even a single object causes both positive and negative signal values, and a collection of objects located in relative proximity to each other generates signal waves that, overlapping each other, make it difficult to identify the original objects correctly. Figure 23 shows a cross-section in the vertical plane through an example sample illustrating the mutual disturbance of waves in the GPR image.

Therefore, the following heuristic algorithm is proposed to eliminate these problems and identify primary objects in the 3D image of the GPR-derived signal:Data from GeoRadar d(x,y,z) have been taken (Figure 22).A certain level of blur *b* was taken, and then the data in each *z*-plane according to the formula was averaged:
(5)$$\underset{z\in (z_{d},z_{u})}{⋀}d_{blur}\left(x_{i},y_{j},z\right)=\frac{1}{b^{2}}\sum_{x=x_{i}}^{x_{i+b}}\sum_{y=y_{j}}^{y_{j+b}}d\left(x,y,z\right)$$After this procedure, the data matrix is shown in Figure 24b and Figure 25b.For each plane $z\in (z_{d},z_{u})$, a histogram *H* from the matrix $d_{blur}\left(x_{blur},y_{blur}\right)$ was created.For each of the planes $z\in (z_{d},z_{u})$, it was decided whether and how the data from it for further calculations, taking into account the significance level *t*, was taken. In this way, the planes *z* in which signals are smaller or larger than the remaining background noise (Figure 23) were found. Analyzing the histogram data *H*, it was verified which condition W is met in Table 4 below.For all planes $z\in (z_{d},z_{u})$, we count the points $\left(x_{blur},y_{blur}\right)$ where the values selected above occurred, assigning a value of 1.0 to each of them regardless of the original signal value at that point. Then, selected points were summed up over the height of the entire data sample, obtaining a two-dimensional matrix $d_{all}\left(x_{blur},y_{blur}\right)$ (Figure 24c and Figure 25c).A histogram for the two-dimensional matrix $d_{all}\left(x_{blur},y_{blur}\right)$ was created, and the *m* largest values were considered consecutively. For each of them, the values of the matrix $d_{all}\left(x_{blur},y_{blur}\right)$ occurring in its neighborhood were analyzed. Let the pair of numbers $\left(x_{j},y_{j}\right)$ stand for the coordinates of the *j*-th point (j=1,2,…,m). To determine the object’s shape causing the signal disturbance at a given location, the coefficient was calculated according to Table 5 below.The largest of the *S* coefficients calculated in Step 6 was selected, thus denoting the shape of the object causing the analyzed disturbance. The calculations in Step 6 were performed for each of the n largest analyzed values in the matrix $d_{all}\left(x_{blur},y_{blur}\right)$ (Figure 24c and Figure 25c).

It should be noted that the algorithm proposed above has several independent parameters:Blur level *b*;Significance level *t*;Signal level *m*;Predicted number of disturbances *n*;Potential shapes of objects we want to recognize *s*.

These parameters are selected each time by trial and error on test data so that primary objects are recognized to the best possible extent. Then, these parameters are applied to the entire set of available data. Their values will depend each time on the power of the measuring device, the size of the sample, and the size, number, and shape of the desired objects. For the source data analyzed, the blurring degree b=5, the significance level t=0.3, the signal level x=7, the number of perturbations n=5, and the potential possibility of all the shapes in Table 5 were analyzed. After analyzing the data for Board B with Defect 1 and Board C with Defect 2, the results shown in Figure 24 and Figure 25 were obtained.

It can be observed that for both samples analyzed, the proposed algorithm allows finding and recognizing the shape of most of the objects generating signal disturbances. To a large extent, the recognized positions of the objects overlapped with the original ones. In the case of Board C with Defect 2, only one object—a plastic tube placed along the y=x axis—was completely unrecognized. On the other hand, the positions of several objects were recognized with some inaccuracy. In further research, the authors will improve the measurement methods to obtain more accurate GPR data and develop the proposed algorithm.

## 4. Conclusions

This paper presented different approaches to the classification of CIPP defects in prepared test specimens in the shape of the boards. The boards were manufactured with artificially induced delamination defects. Additionally, the boards were placed on a sand bed with simulated caverns. The objective of the paper was to assess whether the selected non-destructive methods are capable of classifying both delamination within the test boards and caverns beneath the test specimens. These defects were simulated to represent real-life scenarios where uneven temperature gradients, resulting in the uneven curing of polymer, may lead to delamination. Moreover, if delamination has occurred in the past, it may have formed a cavern between the host pipe and the CIPP. The diagnostic approach could deal with these defects in two main scenarios:Directly after the retrofit action to make a quality assessment;When an already retrofitted pipe needs to be repaired or evaluated.

The IE tests demonstrate that internal defects within CIPP can be localized and assessed using ML or DL approaches. In total, 1302 signals were measured from all 4 test specimens. Internal defects, such as delamination, can be identified in the measured IE signals with 95% accuracy in the presented test specimens using DL; the cavern class reaches an accuracy of 78%, and the matrix class reaches an accuracy of 0.95% in the presented testing specimens. Typically, defects behind the CIPP are retrofitted by injecting epoxy polymers between the host pipe and the CIPP. This process is often based on workers’ experience. Characterizing the CIPP state can be achieved by injecting at the exact spot, resulting in the use of less material and a higher likelihood of extending the life expectancy. The limitation of the IE method lies in the need for mechanical manipulation of the impacting hammer and receiving microphone, which need to move from point to point. Different research groups solve this problem using automated actuators or robot arms [69]. Also, this method in the presented setup cannot operate underwater.

For the detection of larger caverns or pipe intakes above buildings, GPR outperforms IE. The majority of currently manufactured compact ground-penetrating radars are specifically designed for localizing re-bars, cracks, or cables. In our paper, we introduced an algorithm tailored for locating air caverns behind CIPP materials, showcasing a remarkable accuracy of 82.5%. In comparison to IE, GPR achieves a higher accuracy in localizing caverns. While the overall technical requirements for utilizing ground-penetrating radar surpass those of impact-echo testing, GPR can be applied in linear structures, akin to laser-scanning heads. IE measurements necessitate testing at mesh points, which, from an automated perspective, is more intricate than utilizing a static moving radar antenna. The GPR method is effective on plastic or concrete host pipes but faces challenges within metal pipes or in cases of monolithic concrete surrounding the host pipe. In such scenarios, the IE method yields better results due to the reflection of GPR radio waves from the metal host pipe.

Impedance spectroscopy could be employed for relatively fast and continuous measurements, where a continuous signal can be acquired. That characteristic makes this method a promising supplementary approach for characterizing CIPP defects in situ. However, more testing is needed to better understand the dependency of the dielectric constant on the CIPP material state compared to the other two approaches.

It is important to note that the results presented in the paper are still of a laboratory nature and serve as a foundation for further research. As this research progresses, the procedures presented in the paper are planned to be transited to a real-world setting. Consequently, testing in a polygon that closely mimics the real environment will proceed. The insights gained from both laboratory and polygon testing will inform the application of these procedures to a real pipeline repaired using CIPP technology.

## Figures and Tables

**Figure 1 materials-16-07570-f001:**
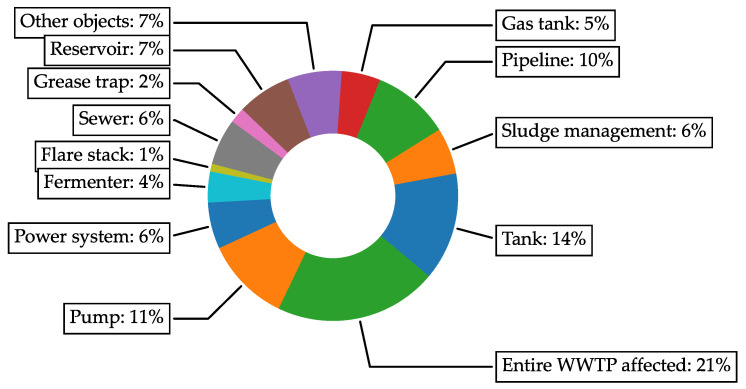
Comparison of wastewater treatment plants accidents in Europe (Adapted from [4], with permission from Elsevier).

**Figure 2 materials-16-07570-f002:**
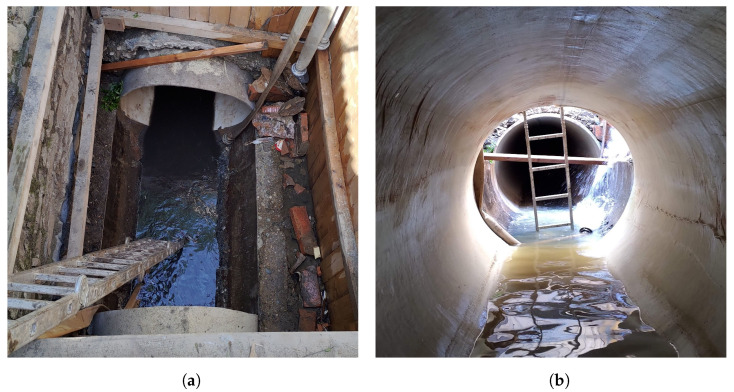
Example of in situ retrofitting action from Brno Lužánky park: (**a**) Image of the digged starting staff; (**b**) Image from inside the newly cured pipe in the old, damaged sewage pipe. Images taken by Prof. Luboš Pazdera.

**Figure 3 materials-16-07570-f003:**
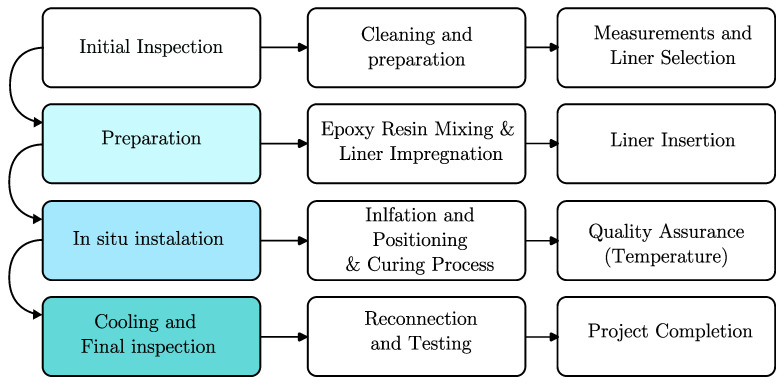
The process of retrofitting action using cured-in-place pipe concept.

**Figure 4 materials-16-07570-f004:**
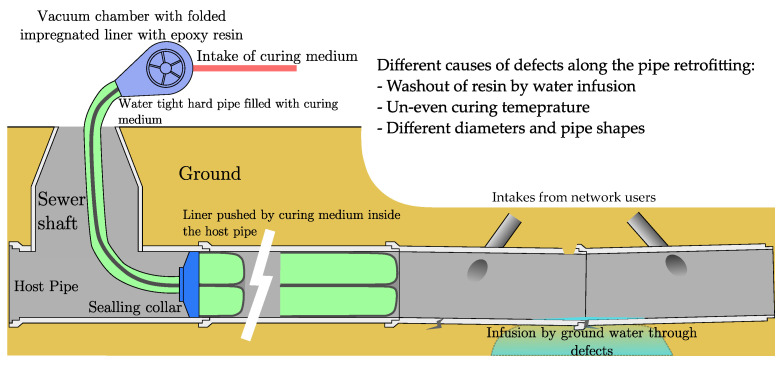
Illustration of CIPP retrofitting process.

**Figure 5 materials-16-07570-f005:**
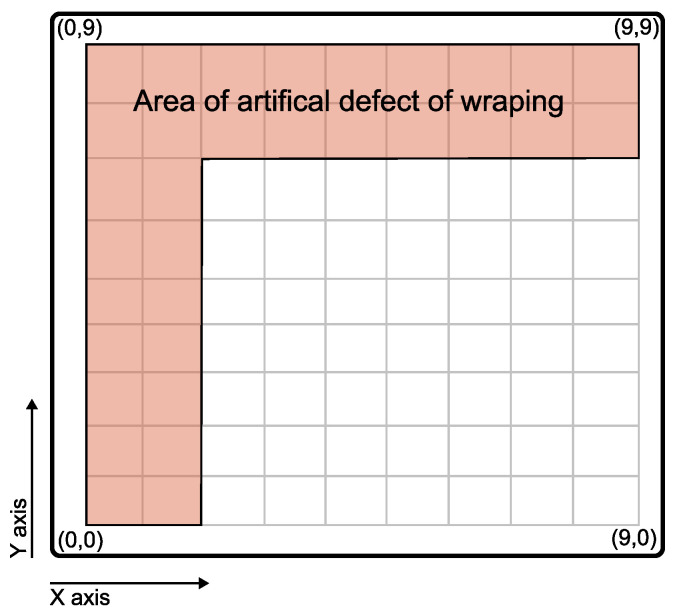
Illustration of artificial wrapping defects on manufactured boards.

**Figure 6 materials-16-07570-f006:**
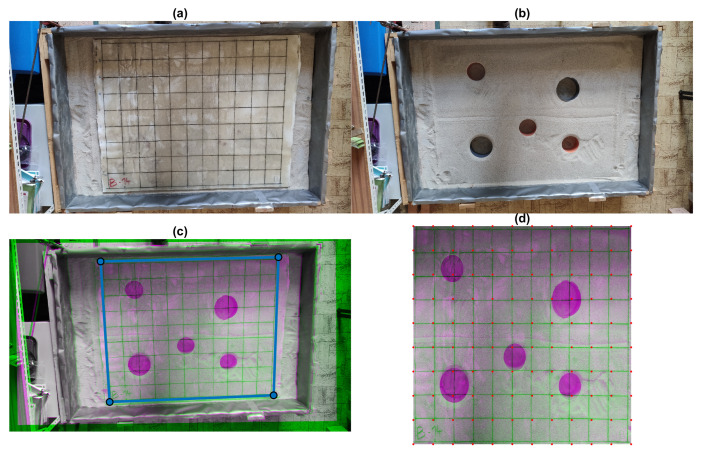
Generation of XY testing point coordinates: (**a**) Board B on sand bed; (**b**) Defects in sand bed; (**c**) Registered image of defects and board; (**d**) Final transformed XY testing point mesh.

**Figure 7 materials-16-07570-f007:**
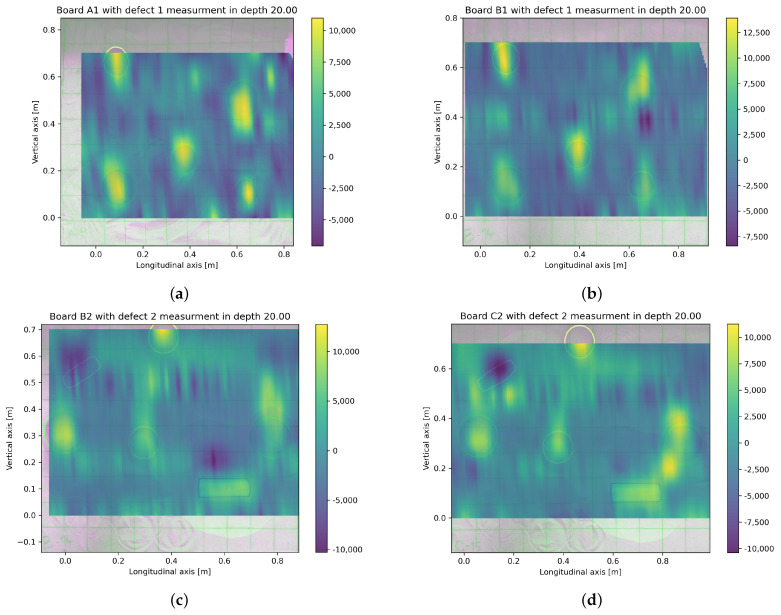
Example of defects in the sand bed: (**a**) Board A1 green pipe ⌀ 150 mm, yellow pipe ⌀ 100 mm; (**b**) Board B1 green pipe ⌀ 150 mm, yellow pipe ⌀ 100 mm; (**c**) Board B2 green pipe ⌀ 150 mm, yellow pipe ⌀ 100 mm, blue metal can, white plastic bottle; (**d**) Board C2 green pipe ⌀ 150 mm, yellow pipe ⌀ 100 mm, blue metal can, white plastic bottle.

**Figure 8 materials-16-07570-f008:**
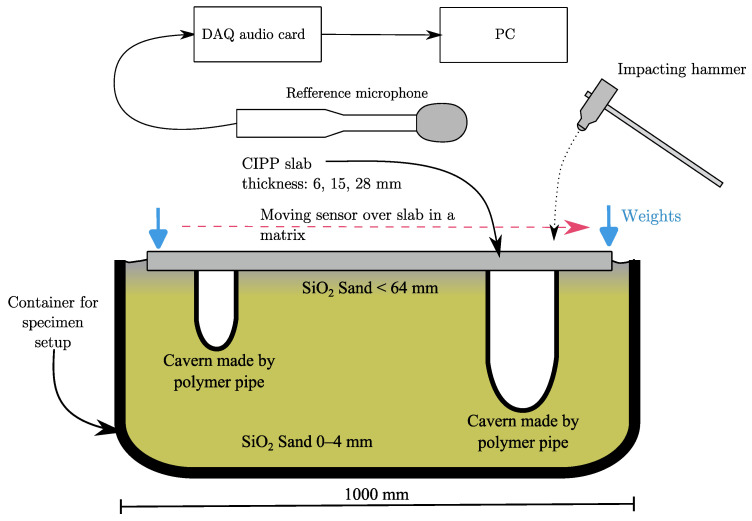
Illustration of IE experiment setup.

**Figure 9 materials-16-07570-f009:**
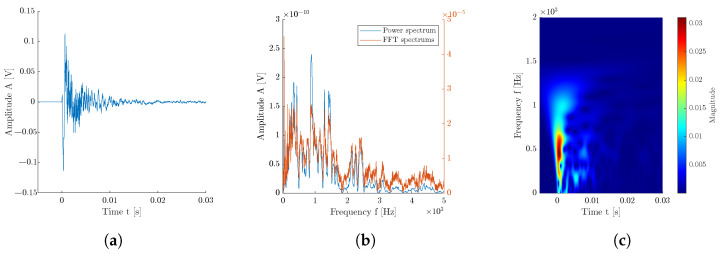
Example of a signal from class ‘Delamination’: (**a**) Measured signal; (**b**) Comparison of spectrum acquired by using FFT and by computing power spectrum; (**c**) Scalogram obtained by using continuous wavelet transformation.

**Figure 10 materials-16-07570-f010:**
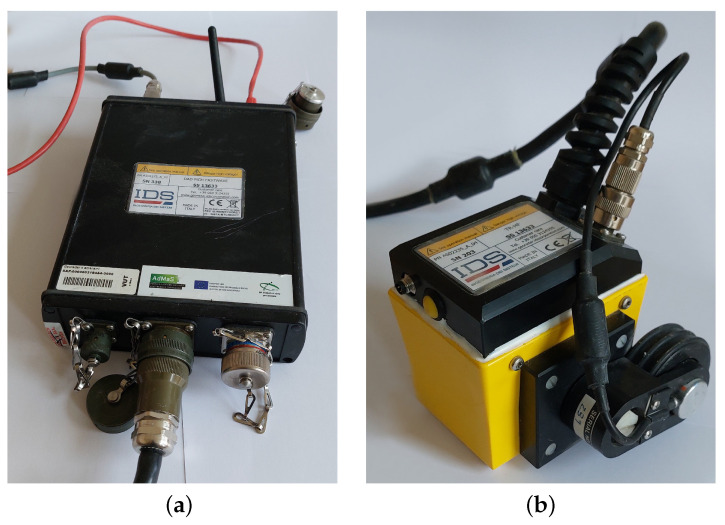
Used GPR for the measurements: (**a**) Measuring card; (**b**) Measuring probe with antenna and tracking wheel.

**Figure 11 materials-16-07570-f011:**
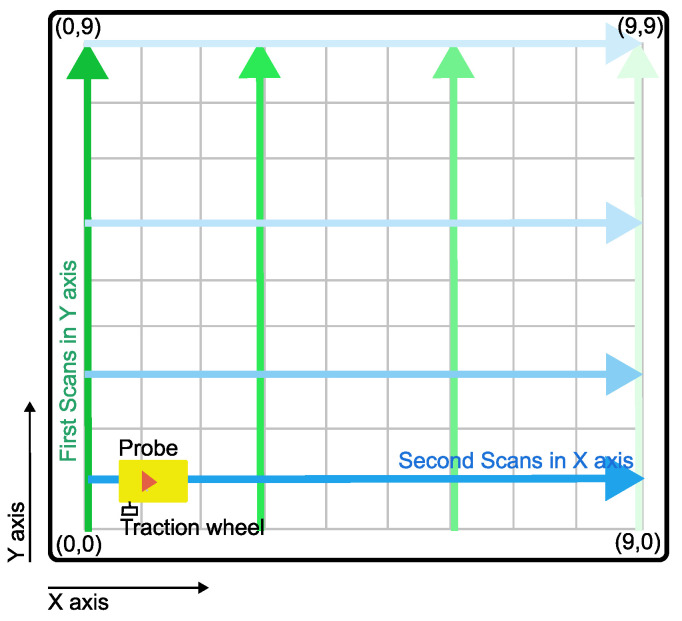
Procedure of volumetric scanning with used georadar. First scans were done in Y axis, and subsequent scans were done to the right (green color). Second set of scans were don in X axis and subsequent scans were taken from bottom up (blue color).

**Figure 12 materials-16-07570-f012:**
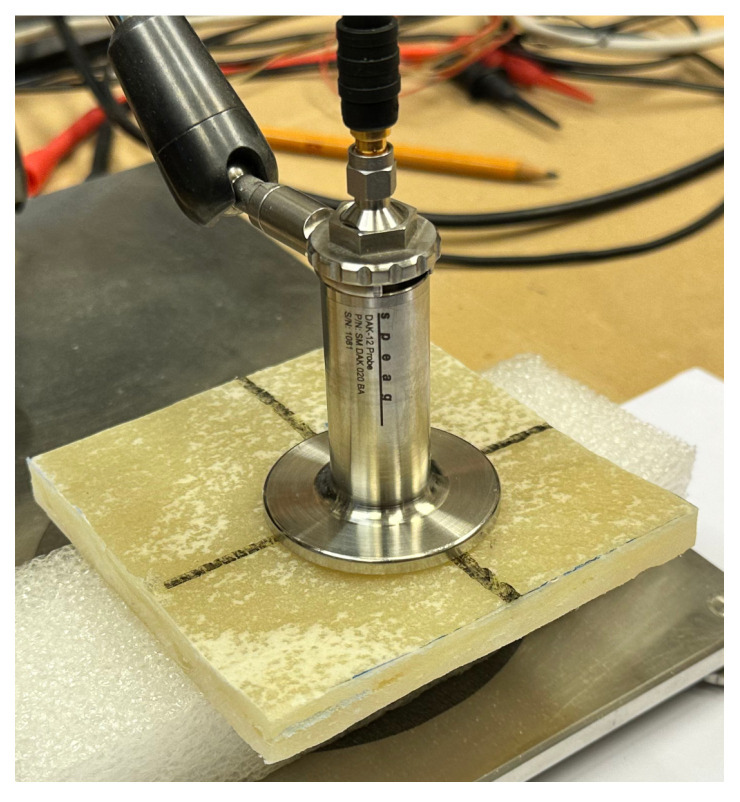
Impedance probe DAK-12 used on tested board pieces. Image taken by Richard Dvořák.

**Figure 13 materials-16-07570-f013:**
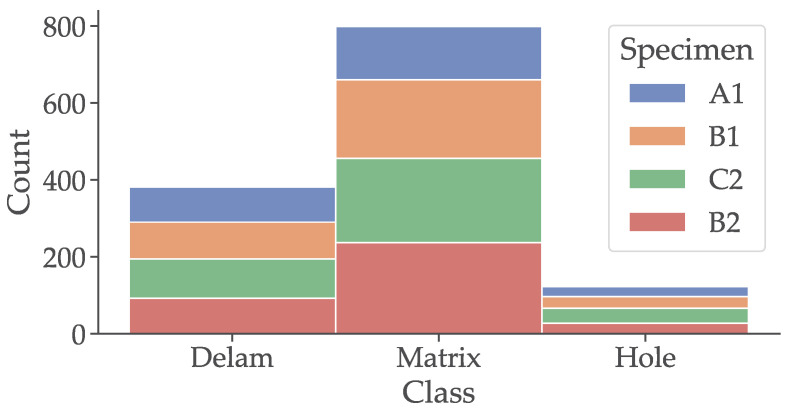
Comparison of all acquired signals and their distribution across the selected classes.

**Figure 14 materials-16-07570-f014:**
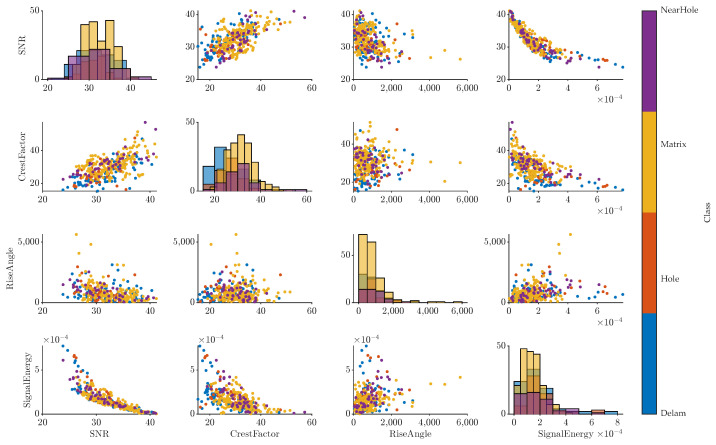
Correlation plot of selected variables of measured signals: signal-to-noise ratio; crest factor; rise angle; signal energy.

**Figure 15 materials-16-07570-f015:**
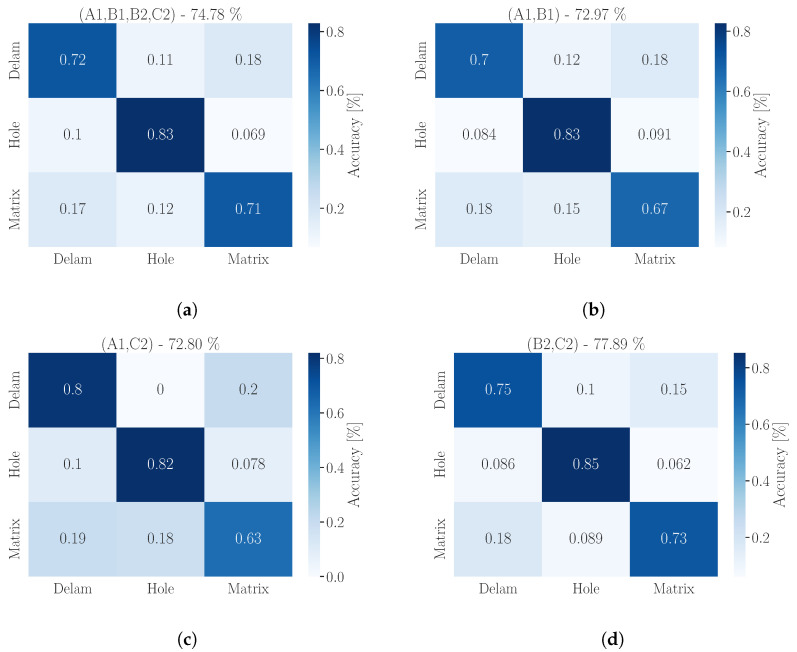
Comparison of classification model accuracy for 4 groups: (**a**) All specimens (A1, B1, B2, C2); (**b**) Only specimens with same defects (A1, B1) and (**c**) (B2, C2); (**d**) Only specimens with different defects (A1, C2).

**Figure 16 materials-16-07570-f016:**
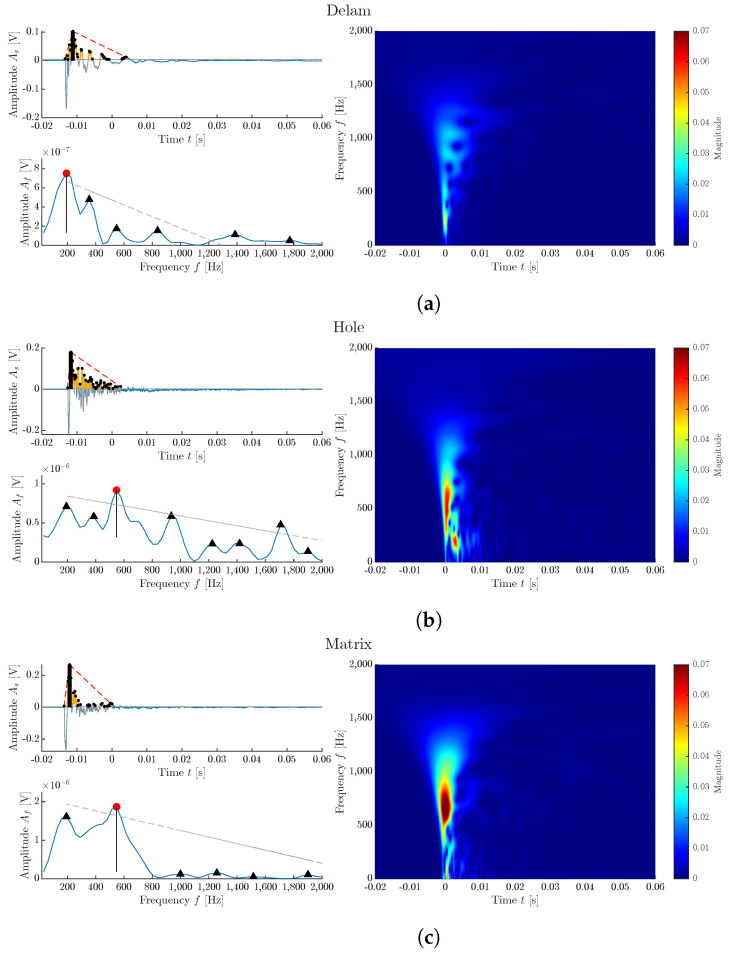
Comparison of representative signals of each Class: (**a**) Delam; (**b**) Hole; (**c**) Matrix. The red dots represent the dominant frequency of the spectrum, black triangles represents the total hits above the threshold value in signal analysis and other dominant frequencies in frequency spectrum. The Red line showing rise time of the signal, and total duration of the signal. Grey line represents dominant linear trend of other frequencies apart the dominant frequency.

**Figure 17 materials-16-07570-f017:**
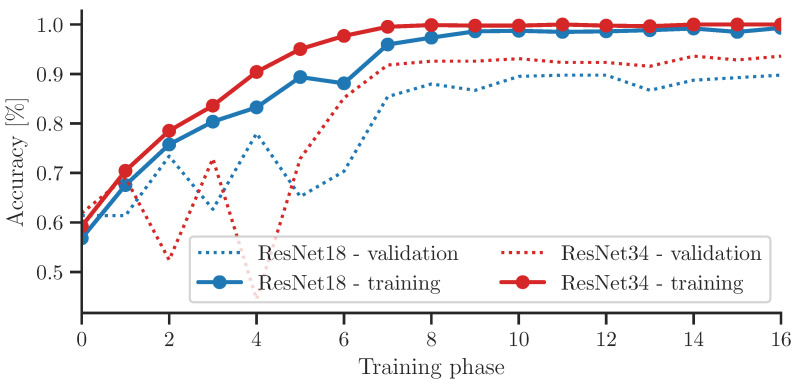
Learning rate of used networks.

**Figure 18 materials-16-07570-f018:**
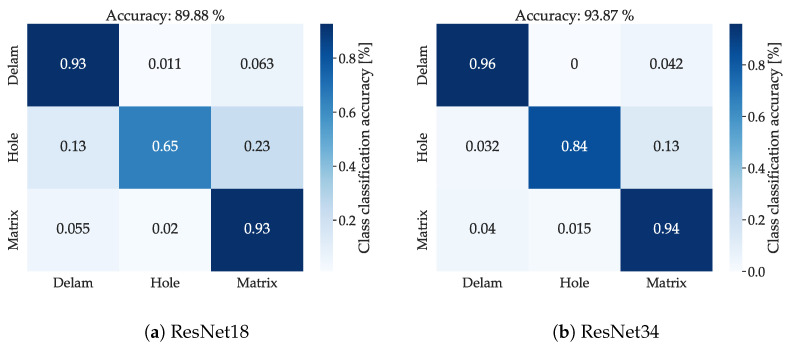
Confusion matrix of each of the used CNN for classification of IE data: (**a**) Network ResNet18; (**b**) Network ResNet34.

**Figure 19 materials-16-07570-f019:**
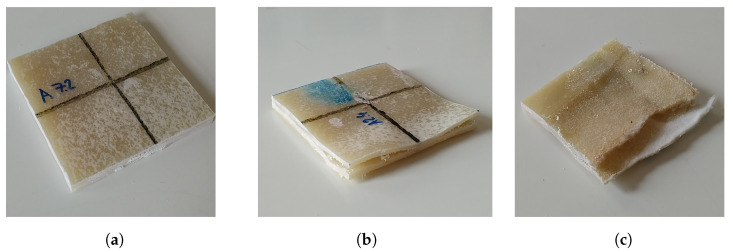
Cut-out specimens from Board A: (**a**) “Healthy” specimen without visible delamination; (**b**) Specimen with present delamination; (**c**) Specimen with highest delamination, also with wash-out epoxy resin. Images taken by Jan Puchýř.

**Figure 20 materials-16-07570-f020:**
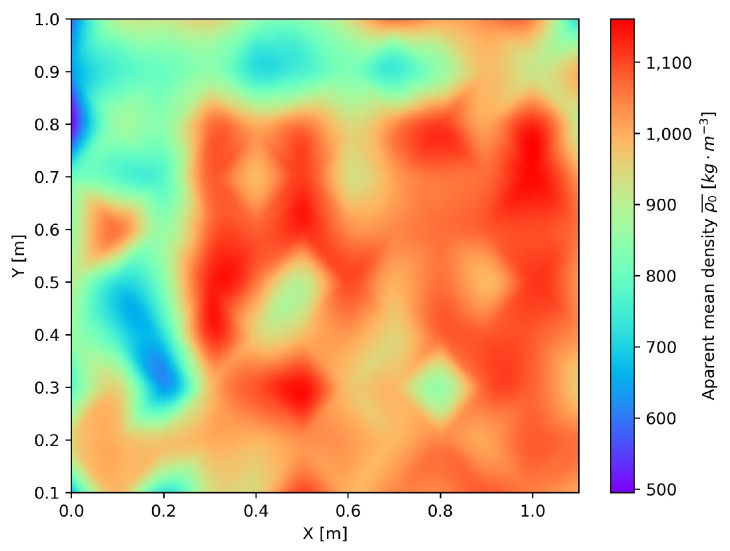
Apparent density of cut-out specimens of Board A.

**Figure 21 materials-16-07570-f021:**
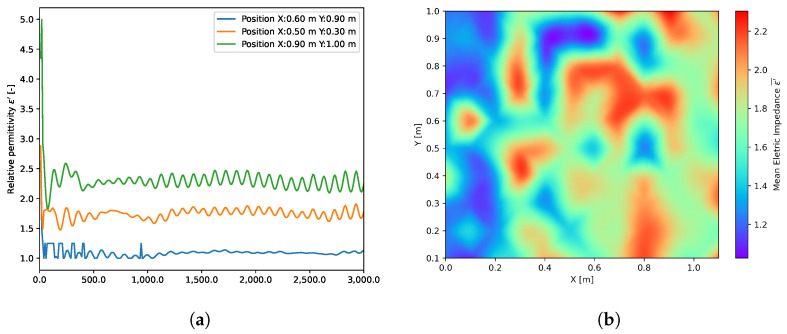
Results of impedance measurement: (**a**) Example of lowest (blue), middle (orange), and highest measured impedance (green); (**b**) Heat map of the distribution of impedance across the board.

**Figure 22 materials-16-07570-f022:**
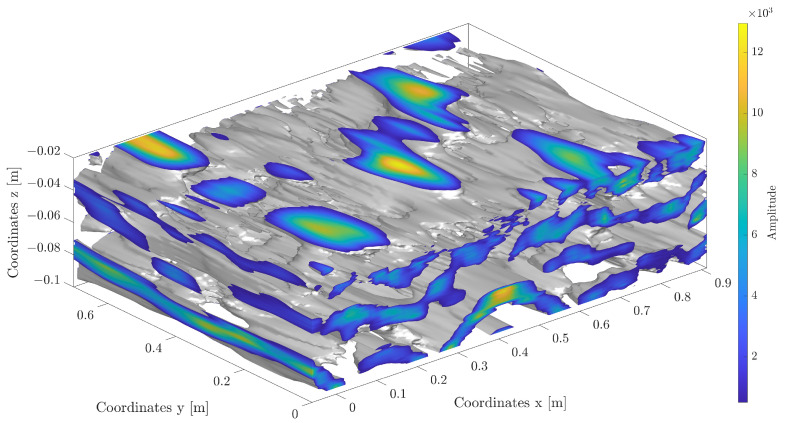
Volumetric scan of Board B with defects type 1.

**Figure 23 materials-16-07570-f023:**
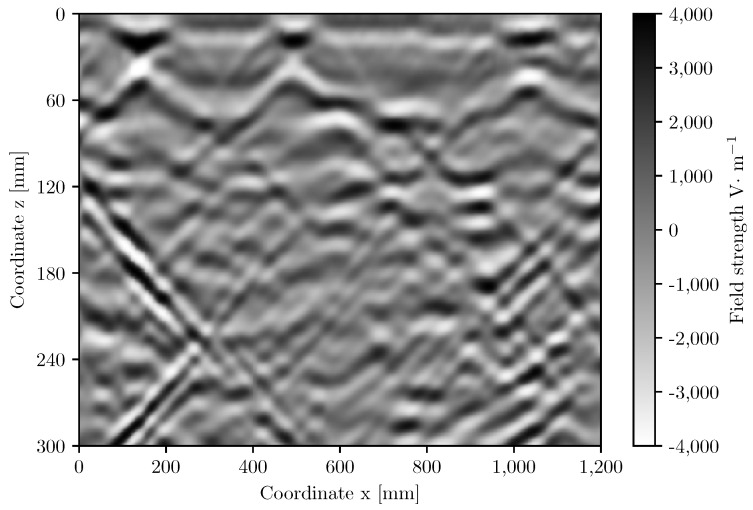
Distribution of GPR signal waves on the vertical cross-section of the sample.

**Figure 24 materials-16-07570-f024:**
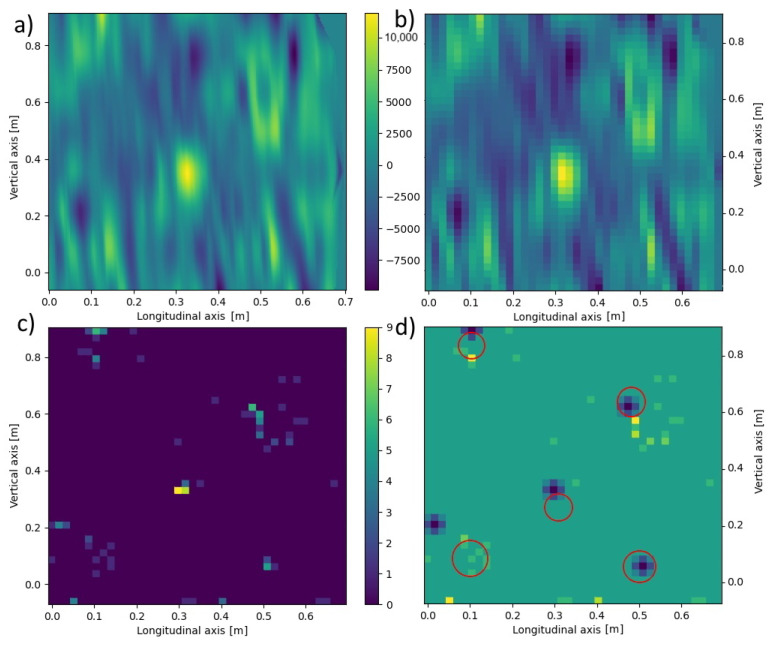
Sequential steps of the algorithm that recognize objects on the data for Board B with Defect 1 from GPR: (**a**) Raw data for z=0.70 m; (**b**) Averaged data for dla z=0.70 m and b=5; (**c**) Matrix $d_{all}\left(x_{blur},y_{blur}\right)$ matrix obtained for n=7 and t=0.3; (**d**) Recognized shapes (blue) versus originally found in the sample (red).

**Figure 25 materials-16-07570-f025:**
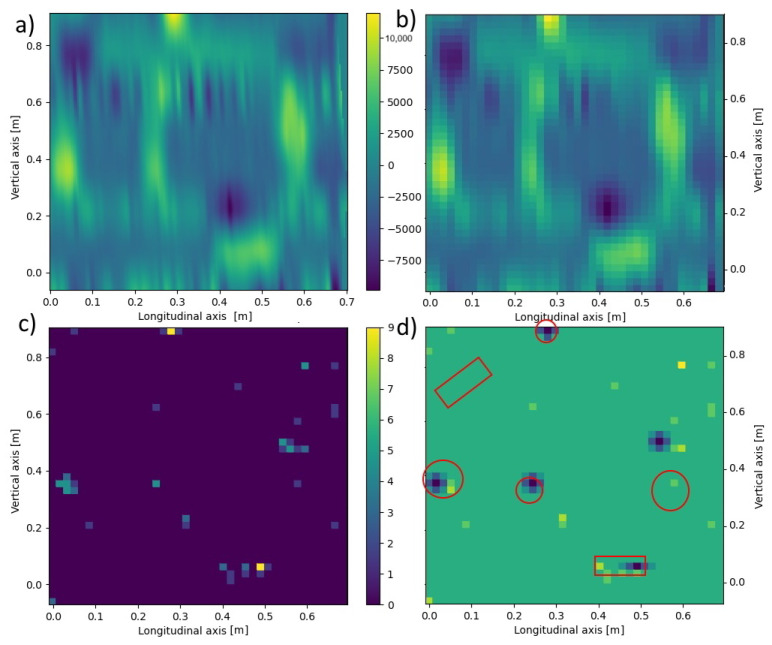
Sequential steps of the algorithm that recognize objects on the data for Board C with Defect 2 from GPR: (**a**) Raw data for z=0.70 m; (**b**) Averaged data for dla z=0.70 m and b=5; (**c**) Matrix $d_{all}\left(x_{blur},y_{blur}\right)$ obtained for n=7 and t=0.3; (**d**) Recognized shapes (blue) versus originally found in the sample (red).

**Table 1 materials-16-07570-t001:** Parameters of CIPP manufacturing process.

Parameter	Description	Type
Resin	Epoxy	IN-EPOX 6040
Fabric	PES non-woven fabric	PES Felt PPX
Curing temperature	20 °C	

**Table 2 materials-16-07570-t002:** Board and defect setup.

Board Name	Defect Set	Thickness (mm)	Number of Liners	Specimens Taken
A	1	8.6	2	x
B	1	14	3	
B	2	14	3	
C	2	15	3	

**Table 3 materials-16-07570-t003:** Parameters of used training setup computer for training ANN.

Parameter	Description
Processor	Intel i5-10600 3.30 GHz
Memory	64 GB DIMM
GPU	NVIDIA Quadro RTX 4000
Input image	652×514×3
PyTorch library	2.0.2 + cu117

**Table 4 materials-16-07570-t004:** The method of considering the significance *t* factor.

*W* Condition	Data for Further Analysis
max(H(z)))>|min(H(z))|(1+t)	The n largest values from the histogram *H*
|min(H(z)))|>max(H(z))(1+t)	The n smallest values from the histogram *H*
In all other cases	No significant data in the analyzed *z* level

**Table 5 materials-16-07570-t005:** Coefficient *S* values for different predicted shapes of objects.

Shape	*S*
□	$\sum_{k=-1}^{1}\sum_{l=-1}^{1}d_{all}\left(x_{j+k},y_{j+l}\right)$
○along the *x*-axis	$\sum_{k=-3}^{3}d_{all}\left(x_{j+k},y_{j}\right)$
○along the *y*-axis	$\sum_{k=-3}^{3}d_{all}\left(x_{j},y_{j+k}\right)$
○along the y=x axis	$\sum_{k=-3}^{3}d_{all}\left(x_{j-k},y_{j+k}\right)$
○along the y=−x axis	$\sum_{k=-3}^{3}d_{all}\left(x_{j+k},y_{j-k}\right)$

## Data Availability

All measured data are available from the authors upon request.

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
