# Peer review of "Non-Destructive Characterization of Cured-in-Place Pipe Defects"

_materials, 2023, doi:10.3390/ma16247570_

Round 1

Reviewer 1 Report

Comments and Suggestions for Authors

In this manuscript, some observations are interesting, but they are rather limited and do not advance the subject. Furthermore, the manuscript has some formatting and grammar problems. Some specific comments are listed below:

(1)   It is recommended to reedit Fig. 4, some fonts in the figure are not clear.

(2)   When dividing the test and training sets, why do the authors use a ratio of 90/10 instead of 80/10 or 75/25? Does the chosen ratio have an impact on the results?

(3)   Line 341, “The used ground penetrating radar illustration is shown on 10” should be corrected to The used ground penetrating radar illustration is shown on Fig. 10”.

(4)   The abstract show that this study aims to compare traditional machine learning and deep learning methods to characterise selected CIPP 13, there is no obvious comparative information in this manuscript, the authors should strengthen the explanation of this part.

(5)   The language of this manuscript needs to be refined, and some of the expressions are unprofessional.

(6)   There are some errors in Fig. 17, the lines corresponding to training and validation do not appear in the figure.

(7)   The authors can cite the below paper for reference of method contribution:

[1] Kang, Q., Chen, E.J., Li, Z.-C., Luo, H.-B., Liu, Y., Attention-based LSTM predictive model for the attitude and position of shield machine in tunneling. Underground Space 2023, 13, 335-350.

Comments on the Quality of English Language

 The language of this manuscript needs to be refined, and some of the expressions are unprofessional.

Author Response

Dear Reviewer 1:

We appreciate your time and dedication, and thank You for Your review. We have worked on all remarks, and worked them in our paper. Here are our responses:

(1) It is recommended to reedit Fig. 4, some fonts in the figure are not clear.

We Have make the font two time larger. The image is in vectors, and because the paper will be only in the electronic version, the font will maintain its resolution in any given zoom.

(2) When dividing the test and training sets, why do the authors use a ratio of 90/10 instead of 80/10 or 75/25? Does the chosen ratio have an impact on the results?

We have used for the ML on the signal features method k-fold cross validation, because for the selection of the optimal model 22 different models were tested. For the DL We have initially choose ratio of 60% for training dataset, 35% for validation dataset and 5% for testing dataset, but because the testing on 5% showed almost 100% accuracy, we have sticked to the ratio 75:25 for training and testing dataset. Thank you for this question, I have added explanation and put some clarity into training, validation and testing groups description, where validation and testing sometimes misinterprets as the same group.

(3) Line 341, “The used ground penetrating radar illustration is shown on 10” should be corrected to “The used ground penetrating radar illustration is shown on Fig. 10”.

We have corrected this part.

(4) The abstract show that this study aims to compare traditional machine learning and deep learning methods to characterise selected CIPP 13, there is no obvious comparative information in this manuscript, the authors should strengthen the explanation of this part.

We have added confusion matrices for ML to be comparable with confusion matrix from DL. By this approach a accuracy on each classes and two distinct approaches can be observed. We have commented the upsides and downsides of both approaches. Thank you for this question.

(5) The language of this manuscript needs to be refined, and some of the expressions are unprofessional.

We have asked for the review of the native speaker.

(6) There are some errors in Fig. 17, the lines corresponding to training and validation do not appear in the figure.

Initially, the figure had gray lines showing, dashed line represent validation curve and full pointed line represent training curve, and colors represent ResNet18 and ResNet34, but it was not clear. I have changed it, so all curves has clearly their labels.

(7) The authors can cite the below paper for reference of method contribution: [1] Kang, Q., Chen, E.J., Li, Z.-C., Luo, H.-B., Liu, Y., Attention-based LSTM predictive model for the attitude and position of shield machine in tunneling. Underground Space 2023, 13, 335-350.

I have cited the mentioned paper in section 2.5. Machine learning.

Once again, I thank you for your time and the review, I believe it make the paper more accurate and understandable.

Yours Sincerely,

Team of authors

Reviewer 2 Report

Comments and Suggestions for Authors

The study presented in this manuscript, regarding the nondestructive characterization of CIPP defects, is interesting and provides useful and information to the field. Congratulations to the authors for their work. However, some aspect could be improved to make the manuscript completer and more comprehensible. 

Mayor comments

1.     Abstract: a brief description of the main results should be included in the abstract.

2.     Introduction: very few references support the statements made by the authors along this first section. Citing one reference at the end of each paragraph is certainly not enough. Please look for specific scientific references supporting the statements made, and locate them beside the corresponding sentence in each case.

3.     Section 2: the experimental setup and the testing techniques are well defined, but a description and justification of the overall methodology would be necessary (why these specific tests and not other, in which order are they applied, on which samples each kind of test, how many samples for test…).

4.     Section 2.2 and 2.3: it would be interesting to explain and give some references about previous and common uses of IE and GPR

5.     Section 2.5: first three paragraphs (lines 377 to 388) are too general and provide few useful information to the article. The rest of the section provides also mainly general definitions and concepts, please focus on the description of the particular aspects of machine learning that are used in the study. 

6.     Conclusions: this section would need a revision, better highlighting the main results obtained in the study (including values) and the validity and limitations of each technique for detecting defects in CIPP.

Minor comments

1.     Addresses of authors #2 and #3 seem to be swapped.  

2.     Abstract: the first statement of the abstract is too strong. Maybe at least changing from “constitute thefoundational infrastructure” to “constitute a foundational infrastructure” would be recommended.

3.     Abstract: CCTV acronym is not defined. Please, avoid as far as possible the use of acronyms in the abstract.

4.     Keywords: although up to 10 keywords are allowed in this journal, I would recommend choosing the most relevant 5-6 keywords and deleting the rest.

5.     Some acronyms are defined more than once, for example NDT is defined in lines 91, 187 and 261; or CIPP in lines 67 and 133. Please define each acronym only once, the first time it is used in the article (excluding the abstract).

6.     Figure 6: it is not necessary to indicate the author of the photographs if he/she is one of the authors of the study.

7.     Equation (1): the variables used in the equation has not been defined.

8.     Conclusions. Lines 597-600. Please rewrite the sentence “These defects occur during the retrofit actions in practice daily and mainly originate from the environmental conditions due to uneven temperature gradients resulting in the uneven curing of polymer.”, it is a bit difficult to follow.

9.     Conclusions. Lines 597-600. Please rewrite the sentence “The internal defects within the CIPP, such as delamination, can recognized from the measured IE signals with 95% in the presented testing specimens” to make it more comprehensible.

Comments on the Quality of English Language

English would benefit from a revision.

Author Response

Dear Reviewer,

Please find our responses in the attachment below.

Sincerely, The Team of Authors

Round 2

Reviewer 1 Report

Comments and Suggestions for Authors

The revision is generally satisfactory. The manuscript can now be accepted for publication.

Reviewer 2 Report

Comments and Suggestions for Authors

Thank you for your answer and your work on the manuscript. I think it has substantially improved, and all my comments have been properly answered and implemented in the text.